# SELF-SUPERVISED CONTINUAL LEARNING

**Kartik Thakral[1], Surbhi Mittal[1], Utkarsh Uppal[2], Bharat Giddwani[2], Mayank Vatsa[1], Richa Singh[1]**
[1]IIT Jodhpur, India [2]NVIDIA
{thakral.1, mittal.5, mvatsa, richa}@iitj.ac.in
uppalutkarsh@ieee.org, bgiddwani@nvidia.com

## ABSTRACT

This paper proposes *Self-Supervised Continual Learning (SCL)* for regularization-based class incremental learning. The novel pretext task in SCL utilizes randomly-transformed labels without depending on data-augmented transforms. *SCL* trained with a novel incremental task-regularizer and an orthogonal weight modifications backbone shows promising performance on three datasets.

## 1 INTRODUCTION

Humans have an unparalleled capacity to incrementally apprehend newer tasks in their daily lives without forgetting the skills gained in the past. Though deep learning models show plasticity towards learning new tasks, unlike humans, they suffer from catastrophic forgetting (McCloskey & Cohen, 1989; Ratcliff, 1990; McClelland et al., 1995; French, 1999). These models optimize their performance based on the objective function and only extract features that are necessary for the current task. This can lead to a loss of prior information, as the model has no idea of the features necessary for joint classification with prior learned tasks. For instance, if a model learns to classify a dog and a bird by their leg count, it may not have enough information to classify a cat in the next task. Without memory replay, the model is unable to mine extra information about previous classes as they are not available anymore. As more tasks are added in the continual setting, the model selects only the features necessary for the current incremental task, which can lead to missing parts affecting the joint classification accuracy. To address this problem in a Class Incremental Learning (CIL) setting, we propose a self-supervision-based approach, termed as Self-Supervised Continual Learning (SCL) that is computationally efficient, easy to deploy, and maintains plasticity without additional memory reserves. By utilizing limited data available at each incremental task, we employ self-supervised pretraining termed as *Random Label Augmentation (RLA)* to extract a diverse set of features. The RLA pretext task synthesizes information from different parts of an image using randomly generated labels in a multi-task setting, thereby providing rich and diverse features to the model (refer Figure 1 in the appendix). Further, a task-wise regularizer is employed that prevents information loss during weight sharing between the pretext and downstream tasks. The downstream training utilizes the *Orthogonal Weight Modifications (OWM) (Zeng et al., 2019)* backbone to mitigate catastrophic forgetting.

## 2 METHODOLOGY

The proposed approach is illustrated in Figure 1 (in the appendix), which showcases the pretext and downstream training framework for a single incremental task t. This training is performed iteratively for each incremental task $t \in [1, T]$.

**Self-Supervised Pre-Training:** In Self-Supervised Learning (SSL), the input images are generally augmented during pretext training, which is computationally inefficient. Consequently, we propose *Random Label Augmentation (RLA)*, based on augmenting the makeshift targets instead of images. These labels are randomly generated for $M$ tasks, with $k$ classes each. The training is performed in a multi-task fashion with different fully-connected layers for each $M$ tasks. The major advantage of *RLA* over the existing approaches (e.g. rotation pretext (Komodakis & Gidaris, 2018)) is that the training time and computational resource usage do not dramatically rise owing to an increase in the amount of augmented data (such as $4\times$ in the case of rotation pretext). Self-supervised pre-training in CIL setting is integrated by training the network for a pretext task $p$ at each incremental task $t$ using images available for that task. During each $t \in [1, T]$, we define a pretext model with parameters $(\omega_p^t, \psi_p^t)$ and downstream model with parameters $(\omega_d{}^t, \phi_d{}^t)$ for feature extraction

| Methods | Split-CIFAR10 | Split-CIFAR100 | | | Split-SVHN |
| --- | --- | --- | --- | --- | --- |
| | 5 tasks | 2 tasks | 5 tasks | 10 tasks | 5 tasks |
| Finetuning | $18.81 \pm 0.59$ | $28.43 \pm 1.08$ | $12.34 \pm 0.62$ | $9.56 \pm 0.55$ | $7.58 \pm 0.57$ |
| iCaRL[*] (Rebuffi et al., 2017) | $50.02 \pm 2.04$ | $24.20 \pm 1.60$ | $22.16 \pm 0.86$ | $19.00 \pm 0.36$ | $71.25 \pm 0.67$ |
| DGM[*] (Ostapenko et al., 2019) | $50.53 \pm 0.46$ | $28.23 \pm 0.75$ | $25.43 \pm 0.14$ | $24.09 \pm 0.19$ | $73.01 \pm 0.77$ |
| OWM (Zeng et al., 2019) | $55.71 \pm 0.49$ | $40.30 \pm 0.65$ | $33.17 \pm 0.79$ | $29.86 \pm 0.33$ | $73.50 \pm 0.81$ |
| MUC[*] (Liu et al., 2020) | - | $33.86 \pm 0.72$ | $28.05 \pm 1.22$ | $22.07 \pm 0.90$ | - |
| IL2A[*] (Zhu et al., 2021) | - | $43.29 \pm 0.43$ | $32.63 \pm 0.86$ | $21.45 \pm 0.67$ | - |
| SSRE[*] (Zhu et al., 2022) | - | $41.06 \pm 0.87$ | $\mathbf{36.82 \pm 0.70}$ | $31.35 \pm 1.0$ | - |
| SCL (w/o RLA) (Ours) | $58.48 \pm 0.18$ | $42.97 \pm 0.31$ | $32.40 \pm 0.48$ | $29.81 \pm 1.10$ | $74.25 \pm 0.34$ |
| **SCL (Ours)** | $\mathbf{59.02 \pm 0.09}$ | $\mathbf{43.69 \pm 0.29}$ | $33.77 \pm 0.26$ | $30.52 \pm 0.72$ | $\mathbf{74.94 \pm 0.27}$ |

Table 1: Average test accuracy for the proposed method. The best performance is depicted by **bold** and the second best by underline. [*] depicts that the results for algorithms were computed with the paper's official code repository using the protocols described in this paper.

and classification, respectively. Weights $\omega_p^t$ are transferred to the downstream task $d$ for $t$. $M$ branches of the pretext training are added to the network corresponding to each of the $M$ tasks. This model is trained as a multi-task learning network where meaningful representations are extracted and for each of the $M$ tasks, and $\psi_p^t$ are trained with the cumulative loss incurred from all the tasks. We minimize the cross-entropy loss function in a multi-task fashion between the target vector of randomly generated labels and the predicted vector. In the CIL setting, the model usually learns discriminative features from images that are required for the current incremental task $t$. These features are not discriminative enough as more tasks are introduced to the model. Pre-training the model with unlabeled data drives the model to extract more and more meaningful information from each input. Thus, the proposed pretext task aims at learning generalized diverse features from the limited data, henceforth saving training time and computational resources. After pre-training, weights $(\omega_p^t, \psi_p^t)$ for $t$, weights $\omega_p^t$ are transferred to the downstream model for $(\omega_d^t, \phi_d^t)$.

**Downstream Training in Class-Incremental Setting:** The downstream task in a CIL setting involves learning weights $(\omega_d^t, \phi_d^t)$ of the downstream model for each incremental task $t$. This is the point where catastrophic forgetting begins to manifest. To address this, we propose an incremental-task regularizer coupled with the Orthogonal Weight Modification (Zeng et al., 2019) backbone. In practical instances, the dataset associated with the new incremental task $t$ may belong to an entirely different distribution. Training the existing model on this new, out-of-distribution dataset may lead to excessive modification of $\omega_p^t$ and $\omega_d^t$. Consequently, the final model may fail to generalize on the old incremental tasks $t_{1:t-1}$. To prevent forgetting at each incremental step, we incorporate a regularization term in the calculated loss, ensuring that the weights are not drastically modified after each pre-training and downstream classification cycle. We add weight regularization as $R(\omega_p^t) = \frac{\alpha}{2} \left\| \omega_p^t - \omega_d^{t-1} \right\|_2^2 + \frac{\beta}{2} \left\| \omega_p^t \right\|_2^2$ between $\omega_d^{t-1}$ of the task $t-1$ and $\omega_p^t$ of the current task $t$ to mitigate forgetting at this step. The regularizer $R(.)$ is minimized with $\alpha$ and $\beta$ as hyperparameters. $\alpha$ aggravates the loss, forcing the model not to deviate much from the model trained on the previous task, and $\beta$ handles the induced sparsity on the model.

## 3 RESULTS

We evaluate the proposed algorithm on split-CIFAR10, split-CIFAR100, and split-SVHN datasets. Table 1 illustrates the efficacy of the proposed algorithm when compared with the existing state-of-the-art regularization-based CIL algorithms. We observe that the SCL algorithm outperforms the existing algorithms on split-CIFAR10 and split-SVHN datasets for 5 tasks. For the split-CIFAR100 dataset, SCL achieves state-of-the-performance for 2 tasks and achieves competitive performance for 5 and 10 tasks. For ablation experiments, we evaluate the performance by fine-tuning each incremental task and observe that the model can retain the information from only the last incremental task $T$. Further, the task-regularizer with OWM backbone is added to the framework, which achieves state-of-the-art performance on split-CIFAR10 and split-SVHN datasets. The contribution of RLA in the pipeline is validated by replacing it with popular SSL pretext tasks. The performance reported in Table 2 (in the appendix) highlights the efficacy of RLA.

## 4 CONCLUSION

We introduce a novel data and compute efficient concept of utilizing self-supervised learning for regularization-based CIL. With this work, we also aim to attract researchers toward the potential of label transformations instead of conventional data augmentation for unsupervised learning.

URM STATEMENT

The authors acknowledge that the first author of this work meets the URM criteria of the ICLR 2023 Tiny Papers Track.

ACKNOWLEDGEMENT

Thakral is partially supported by the PMRF Fellowship. Mittal is partially supported by the UGC-Net JRF Fellowship and IBM Fellowship. Vatsa is partially supported through the Swarnajayanti Fellowship.

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

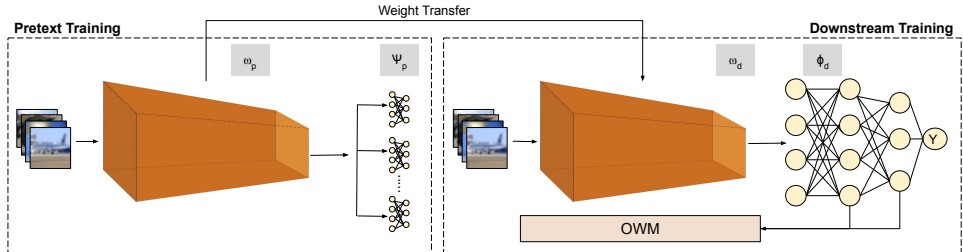

Figure 1: Illustration of the proposed pretext and the downstream task for an incremental task $t$.

## A EXPERIMENTAL SETTINGS

The proposed algorithm is evaluated on three datasets: split-CIFAR10, split-CIFAR100, and split-SVHN. We report the *average test accuracy*, which is defined as the average of test accuracies achieved across all tasks. All experiments are performed using *five seeds*.

### A.1 DATASETS AND PROTOCOL:

The training and evaluation are performed in a class-incremental setting following the standard protocols defined in the works of (Zeng et al., 2019) and Hu et al. (2018) on split-CIFAR10 (Krizhevsky, 2009), split-CIFAR100 (Krizhevsky, 2009), and split-SVHN datasets. The split-CIFAR10 and split-SVHN datasets are used for evaluation on 2 classes per task, and split-CIFAR100 is used for training and evaluation on 10, 20, and 50 classes per task.

| Pretext Task | Average Accuracy (%) | |
|---|---|---|
| | split-CIFAR10 | split-SVHN |
| Rotation (Komodakis & Gidaris, 2018) | 56.70 ± 0.52 | 73.79 ± 0.31 |
| Colorization (Larsson et al., 2017) | 55.82 ± 0.48 | 73.03 ± 0.33 |
| **RLA (proposed)** | **59.02 ± 0.09** | **74.94 ± 0.27** |

Table 2: Performance of the proposed algorithm by replacing the proposed Random Label Augmentation (RLA) task with Rotation and Image Colorization pretext tasks on the split-CIFAR10 and split-SVHN datasets for 5 incremental tasks.

### A.2 COMPARISON ALGORITHMS:

The results of the proposed algorithm are compared with various benchmark algorithms in the domain of regularization-based CIL- (1) OWM (Zeng et al., 2019), (2) MUC (Liu et al., 2020), (3) IL2A (Zhu et al., 2021), and (4) SSRE (Zhu et al., 2022). The baselines are run using the original open-source codes with the same network architecture as the one used in the proposed SCL. The details of this network are described in Section A.3.

### A.3 IMPLEMENTATION DETAILS

For all experiments, we use a 3-layer CNN network consisting of three fully-connected layers. The network architecture is consistent with the one employed in work from Zeng et al. (2019). In our experiments, we train the model using randomly generated labels for three tasks ($M = 3$), each with two classes ($k = 2$). The extracted features are utilized by three separate classification heads, each of which has two fully-connected layers. All training is performed using the Stochastic Gradient Descent (SGD) optimizer for 50 epochs with the learning rate set to 0.001. The hyperparameters $alpha$ and $beta$ have been set to the values 10 and 18 for the split-CIFAR10 dataset and the split-CIFAR100 dataset, respectively, while for the split-SVHN datasets, these values are 5 and 12. All of the experiments are carried out on five seeds, and the results are reported as the overall average, along with the standard deviation for all of the seeds. The algorithm is implemented in Pytorch 1.8. All the experiments are performed on Nvidia DGX station with four V100 GPU cards having 32 GB memory each. The code is publicly available [1].

---

[1]https://github.com/thakral-kartik/Self-Supervised-Continual-Learning

