# OpenReview forum: "Self-Supervised Continual Learning"
_ICLR.cc/2023/TinyPapers — Submitted to Tiny Papers @ ICLR 2023_

### Official Review · Reviewer_uZZY · 2023-03-26

**Confidence:** 5

**Summary Of Contributions:**

This work proposes a self-supervised learning (SSL) phase before each task to improve class-incremental learning (CIL). Particularly, the authors proposed the Random Label Augmentation (RLA) task that randomly assigned labels to the unlabeled data. Experiments showed encouraging results compared to some baselines.

**Rating:**

Great Start (GS): a submission which meets some of the reviewing criteria but has room for improvement

**Strengths And Weaknesses:**

Strengths

- Combining SSL and CIL is a promising and interesting research direction. A suitable pre-text task for CIL will likely to have good impacts to the community.
- The results are encouraging.

Weaknesses
1. My major concern of this work is the motivation of the RLA loss. Particularly, it is unclear to me how RLA can benefit CIL. Training with random labels during pre-text training may cause more harm because the model may need to unlearn the wrong label order during supervised learning. The authors are encouraged to explain the benefits of RLA more clearly.
2. Other concerns and suggestions
* The Self-supervised continual learning setting is unclear. Which data is used during pre-text training, is it the same data during down-stream training?
* The pre-text training phase is essentially a supervised learning. It is unclear how catastrophic forgetting does not arise during this stage.
* Why pre-text training is designed as a multi-task learning instead of a simple single task learning?
* What are the advantages of the proposed RLA compared to contrastive learning (e.g. using SimCLR loss)?
* The claim that RLA has better complexities compared to other baselines may need to be checked more carefully. RLA induces more parameters for the multi-task classifier, which is more expensive compared to the classifier for the rotation pre-text.
* It would be better to implement the proposed framework on better backbones (such as ResNets) which have better representation capability.
* More recent and standard baselines should be included.
* The reference style should be revised.
* The writing can be further improved.



**Suggested Changes:**

Please see **Weaknesses**.

---

> ### Author Response · Authors · 2023-05-24
> **Detailed Response to comments of Reviewer uZZY (Addressing weaknesses and Suggested Changes)**
>
> We thank the reviewer uZZY for their encouragement and constructive feedback. Our response to the weaknesses and suggestions are below:
> 1. The problem with continual learning models is that they do not have the ability to extract multiple meaningful features from images as humans do. They optimize their performance based on the objective function of classification and only extract features that are necessary for the current task. This can lead to a loss of prior information, as the model has no idea of the features necessary for joint classification with prior learned tasks. For example, if a model learns to classify a dog and a bird by counting their legs, it may not have enough information to classify a cat in the next task. Without replaying previous inputs, the model is unable to mine extra information about previous classes because they are not available anymore. As more tasks are added in the continual setting, the model will select only the features necessary for the current and following incremental tasks, which can lead to missing parts affecting the joint classification accuracy. With RLA-based pre-training, we not only allow the model to extract more meaningful features from the same set of images and preserve prior information, we also reduce the training time and resource requirements by augmenting the labels instead of the data. We have added this explanation in the updated draft for better understanding.
>
>
> 2. Other concerns are addressed below:
>
> (2A) For each incremental task $t$, first, a model is trained on the dataset $D^t$ (i.e. the dataset for the incremental task $t$) with randomly augmented labels. After this, the fully-connected layers are dropped, and then the weights of the feature extractor is transferred for the downstream task. This downstream training is performed on the same dataset set $D^t$ with original labels.
>
> (2B) The pre-training phase is self-supervised, where randomly augmented labels are used instead of the original labels. This pretraining helps in extracting meaningful and diverse features instead of features that are optimized only for the current incremental task, assisting in preserving the prior information from different tasks. The pretraining aided with the proposed weight regularizer which preventing catastrophic forgetting by regularizing the updates in weights done in the pretraining phase. This explanation is added in the updated draft.
>
> (2C) By making the pretraining multi-task learning, we enforce the model to extract diverse sets of features to optimize not only for the current task, but for future tasks as well.
>
> (2D) In conventional Self-supervised pretraining like RotNet [1] and SimCLR[2], the data is augmented through some transformations and is assigned with pseudo-labels. This increases the training time and computational resources. The major advantage of the proposed RLA over the existing approaches is that training time and resource usage do not increase significantly as labels are augmented instead of the data. The performance reported in Table 2 also depicts that RLA outperforms the existing self-supervised pretext tasks.
>
> (2E) As pointed out by the reviewer uZZY, we check the time taken for an epoch for both pretext tasks.  We observe that for each epoch, rotation pretext takes 9 seconds, whereas RLA takes only 2 seconds. This is because, in the case of the rotation pretext task, the dataset is scaled to 4 times the original data size, leading to increased training time.
>
> (2F) In order to ensure a fair comparison with existing state-of-the-art techniques such as OWM [3], we used the same backbone network as they did.
>
> (2G) The previous version included the most recent and state-of-the-art algorithms, namely:  OWM, MUC, IL2A, and SSRE. In the revision, we also include some standard baselines for comparison, such as iCaRL and DGM.
>
> (2H) We have updated the references in the revised version.
>
> (2I) In the revised version, we have made editorial revisions to address the reviewer’s feedback.
>
>
>
>
> [1] Gidaris, Spyros et al. “Unsupervised representation learning by predicting image rotations.” International Conference on Learning Representations, 2018.
>
> [2] Chen, Ting, et al. "A simple framework for contrastive learning of visual representations." International conference on machine learning. PMLR, 2020.
>
> [3] Zeng Guanxiong et al. “ Continual learning of context-dependent processing in neural networks.” Nature Machine Intelligence, 2019.

---

### Official Review · Reviewer_zheF · 2023-03-31

**Confidence:** 2

**Summary Of Contributions:**

Authors have provided a way of Self-Supervised Continual Learning (SCL) for regularization based class incremental learning by using randomly transferred labels as opposed to traditional data augmentation techniques.

**Rating:**

Great Start (GS): a submission which meets some of the reviewing criteria but has room for improvement

**Strengths And Weaknesses:**

Strength:
1. Use of randomly transformed labels as compared to general data augmentation is interesting.
2. Author also have shown their method outperformed on split-CIFAR10 and split-SVHN datasets for 5 tasks.

Weakness
1. More information is required for reproducing the results.
2. Better if the paper can be submitted as full paper not as tiny papers, lots of important details will get missed.

**Suggested Changes:**

1. More information is required for reproducing the results.
2. Better if the paper can be submitted as full paper not as tiny papers, along with more details.
3. Better to share codes, so that other researchers can take reproduce the results.

---

> ### Author Response · Authors · 2023-05-24
> **Detailed response to comments of Reviewer zheF**
>
> We thank the reviewer zheF for their encouraging reviews. Our response to the weaknesses and the suggested changes are as follows:
> 1. We performed all the experiments on 5 seeds and have listed all the hyperparameters details in the Implementation Details section of the appendix. We have also updated the draft with details of computational resources utilized for the experiments.
> 2. This paper provides all the details ranging from idea conceptualization, algorithmic details and comparison with state-of-the-art techniques in the main paper. In the appendix, we provide the experimental setup and implementation details, with further experiments to show the efficacy of the proposed algorithm. We try to keep the paper crisp and concise while providing all the necessary details.
> 3. We have shared the codes in the updated draft of the paper.

---

### Meta-Review · Area_Chair_2VXW · 2023-04-05

**Recommendation:** Invite to revise
**Confidence:** 5

**Metareview:**

This paper introduces a new method that integrates self-supervised learning (SSL) and continual learning (CIL), demonstrating promising results on the split-CIFAR10 and split-SVHN datasets. However, as other reviewers have noted, the paper lacks essential details required for reproducibility, making it challenging to replicate the study. Additionally, the paper does not adequately explain the benefits of Random Label Augmentation (RLA) for CIL, raising concerns that training with random labels during pre-text training may be counterproductive.



**Summary:**

The authors introduced Self-Supervised Continual Learning using Random Label Augmentation to improve class-incremental learning.

**Reason For Not Giving A Higher Recommendation:**

The current presentation of the self-supervised continual learning setting is unclear, making it difficult to replicate the paper. Essential details and rationales are not adequately explained, and the claim of a superior technique necessitates further validation. Additionally, the writing quality could benefit from further improvement.

**Reason For Not Giving A Lower Recommendation:**

N/A

---

> ### Author Response · Authors · 2023-05-31
> **Summary of Revisions Made**
>
> We would like to express our gratitude to the reviewers and the meta-reviewer for their valuable feedback and suggestions to improve our paper. In response to their comments, we have made several revisions and would like to highlight the key changes:
>
> 1. We have provided additional details on the implementation and computational resources used in our experiments to enhance reproducibility.
> 2. We have made the code publicly available for reproducibility.
> 3. We have included a more detailed explanation of the motivation behind our proposed pre-text task and why RLA Loss is effective. This will help readers better understand the need for our pre-training approach and its impact on model performance.
> 4. We have also elaborated on how and why our pre-text task is more time and compute-efficient compared to existing techniques.
> 5. In Table 1, we incorporate standard and popular baseline comparisons with existing state-of-the-art methods for a more comprehensive comparison.
> 6. Finally, we have made some editorial changes to improve readability and updated our references as suggested by the reviewers.
>
> We believe that these revisions have improved the quality of our paper and hope that they address the concerns raised by the reviewers.

---

### Decision · Program_Chairs · 2023-04-10

Revision accepted; invite to archive

---

> ### Author Response · Authors · 2023-05-31
> **Opting-in for Archive**
>
> We wish to opt-in for archival (publishing the paper).
>
> Thanks